# Effect of Nitrogen on the Properties of Flax (*Linum usitatissimum* L.) Plants and Fibres

**DOI:** 10.3390/polym14030558

**Published:** 2022-01-29

**Authors:** Ružica Brunšek, Jasminka Butorac, Zvjezdana Augustinović, Milan Pospišil

**Affiliations:** 1University of Zagreb, Faculty of Textile Technology, Department of Materials, Fibres and Textile Testing, Prilaz baruna Filipovića 28a, 10000 Zagreb, Croatia; 2University of Zagreb, Faculty of Agriculture, Department of Field Crops Forage and Grassland, Svetošimunska cesta 25, 10000 Zagreb, Croatia; jbutorac@agr.hr (J.B.); mpospisil@agr.hr (M.P.); 3College of Agriculture at Križevci, Milislava Demerca 1, 48260 Križevci, Croatia; zaugustinovic@vguk.hr

**Keywords:** *Linum usitatissimum* L., cultivars, nitrogen fertilization, morphological properties of plant, physical-mechanical properties of fibre

## Abstract

This paper presents a three-year study of the influence of different amounts of nitrogen on the properties of flax plants and fibres. At the same time, the acclimatization ability of five different cultivars of fibre flax was estimated through the valorisation of their morphological (technical stem length, stem thickness) and physical-mechanical properties of the fibres (length, fineness, tenacity). Cultivar trials with fibre flax were set up across three years (2008–2010) at the following locations: the experimental fields of the Faculty of Agriculture in Zagreb on anthropogenized Eutric Cambisol and the College of Agriculture at Križevci on pseudogley on level terrain. The selected cultivars were fertilized without and with different nitrogen rates (0, 30, 60 and 90 kg/ha) in different time. The trials were carried out according to the RCBD in four replications. According to the results of the three-year study of flax and fibres, significant differences were established among the cultivars and among the added nitrogen rates under study. Based on the results of the morphological and textile-technological properties of flax, the cultivars Viola and Agatha achieved higher values at the location of Križevci, where it was not necessary to add more than 30 kg N/ha.

## 1. Introduction

Flax (*Linum usitatissimum* L.) was one of the earliest ancient plants to be domesticated by humans and grown for seed, oil and the strong fibres produced in its stems [1].

Flax fibres are used as textile raw material, at first for composing cords and weaving yarn, and then for fashionable garments or high-quality fabric upholstery. Over the past decades, its numerous existing applications and the constant development towards more innovative materials make flax a plant of growing industrial interest, especially for the use of flax fibres as composite reinforcement. However, much fewer studies focus on flax as a plant, investigating the relationships between the properties of flax fibres and the development of the plants from which they are obtained. It is necessary to take advantage of the fact that flax is a crop with great potential due to its multiple uses and its high adaptability [2,3,4].

For more than 20 years, there has been cooperation between agronomists and textile-oriented specialists with the aim of revitalizing the sustainable and ecological production of flax fibres in Croatia. Since Croatia does not have its own cultivars [5,6], it is necessary to introduce cultivars from Europe, which are grown in climatic conditions similar to Croatian cultivars [7,8,9,10,11,12]. Therefore, it may be assumed that the newly formed climate conditions would cause the reduction in the valuable properties of flax plant and fibres. Sowing a suitable cultivar is an important factor to enhance growth, yield and the components and quality parameters of flax depending on which area it is planted in with regard to climatic conditions. The applying of appropriate fertilizers at a suitable time and in the required quantities needed for the plant is one of the most essential elements to ensure that the plant provides a high output. Nitrogen fertilization plays an important role in increasing the productivity and quality of flax.

According to our previous two-year study [8,13,14] of the agronomic and morphological properties of fibre flax in two locations, significant differences were established between the cultivars, without and with topdressing by nitrogen. Higher values of the studied properties were obtained with a nitrogen topdressing with 27 kg/ha [14]. According to the obtained results, all investigated varieties showed satisfactory resistance to lodging, regardless of nitrogen fertilization. A large amount of nitrogen increases vegetative flax mass, which causes lodging and increases the stem thickness, and the amount of wooden core decreases the yield and quality of fibres, i.e., further fibre processing and the suitability of flax fibres as a textile raw material for yarn and fabric manufacturing are prevented [8,15,16,17,18]. Phosphorus positively affects the development of the root and fibre strength and takes part in seed formation, while potassium stimulates fibre and seed formation. It creates a stronger bond among fibre bundles, and the stem is technically easier to process [3,8].

Many researchers from Egypt concluded that using nitrogen fertilizer at a suitable and required level could improve the yield, components and quality of flax [19,20,21,22]. This fertilizer is recommended for the highest yield of straw and seed with the best-quality new cultivars, Giza and Sakha, respectively, under the climatic conditions of Egypt. A group of authors from Ethiopia studied the potential development of flax fibre and oil quality (*Linum usitatissimum* L.) under the influence of different rates of nitrogen application. For example, they determined that increasing the nitrogen fertilizer rate increased the flax fibre length. A fibre length of about 58 cm was obtained from a 138 kg N/ha fertilizer rate and the lowest rate was recorded in the control treatment [23].

Bast fibres, including flax fibres, are characteristic of a high level of non-uniform morphology, which can be neither substantially affected nor removed by production technology, but noticeably affects fibre properties. Furthermore, the properties of flax fibres are also the result of the effect of different factors influencing the growth and development of flax plants such as cultivars and agroecological cultivation conditions, as well as fibre extraction from the stem of the flax plant and flax processing into fibres. Closely related parameters, which have an impact on the quality of flax fibres, are ergonomic, morphological and phenological plant properties such as root shape, length, thickness, stem colour, branching, leaf shape and retting procedure [24,25]. It can also be asserted that each step from plant cultivation to fibre production has its own specific impact on fibre quality [3]. Fibre fineness, length and strength are the most important processing properties, determining the quality and suitability of flax fibres as a textile raw material for yarn and fabric manufacturing [2].

There is the assumption that, under different climate conditions, a reduction in the agronomic properties of imported cultivars occurs, in turn causing a reduction in textile-technological fibre properties. Agronomists and textile-oriented specialists from Croatia investigated acclimatization ability of foreign flax cultivars by determining their morphological and phenological flax properties and textile-technological fibre properties [9]. They concluded that the cultivar Viola has a better acclimatization ability and is recommended for cultivation in Croatia. The fibre flax cultivar, Viola, produces fibres of higher quality, which are longer, finer, and have a lower degree of resolution. Therefore, these authors are expanding their research to include a larger number of different cultivars [26,27]. Based on the results of the morphological and textile-technological properties of flax, the cultivars, Agatha, Electra and Viola, achieved higher values, and their fibres satisfy the current market demands for their use. The cultivars fertilized with nitrogen produced taller plants, longer technical stem and fibre lengths, thicker stems, and slightly coarser and stronger fibres. These plants entered the stage of flowering and yellow maturity later than the plants without nitrogen. According to the results obtained for the studied properties of fibre flax fertilization, it is not necessary to add more than 30 kg of nitrogen/ha.

Considering the obtained results, the investigation continued, and the influence of nitrogen fertilization on the agronomic properties (dry stem yield, dry stem yield after retting, total fibre yield, share of total fibre, long fibre yield and share of long fibre) of fibre flax cultivars was investigated [11]. Based on the analysis of the results, it was concluded that the cultivars, Agatha, Viola and Electra, recorded the highest values for the investigated properties (dry stem yield, dry stem yield after retting, total fibre yield, long fibre yield) over three years and the highest share of total fibres and long fibres over one or two years. Additionally, the optimal nitrogen rate for fibre flax, according to the obtained results, was 30 kg N/ha.

This paper presents a continuation of the research on the impact of different amounts of nitrogen on the properties of flax plants and fibres. At the same time, the acclimatization ability of five different cultivars of fibre flax was estimated through the valorisation of their morphological (technical stem length and stem thickness) and physical-mechanical properties (length, fineness and tenacity).

## 2. Materials and Methods

### 2.1. Flax Fibre Growing and Processing

Cultivar trials with foreign fibre flax were carried out in the experimental field of the Faculty of Agriculture of Zagreb (45°49′26″ N, 16°02′07″ E) and in the experimental field of the College of Agriculture of Križevci (46°02′23″ N, 16°54′62″ E). The trials involved five cultivars: Viking (Cooperative Liniere de Fontaine Cany, France), Viola (Van de Bilt Zaden, The Netherlands), Venica (Agritec, Czeck Rep.), Agatha, and Electra (Cebecco Seed, Netherlands). According to type affiliation, the soil in Zagreb is anthropogenized Eutric Cambisol, while in Križevci is it pseudoglay on level terrain. Both soils have silty loamy textures. Because of their high powder content, they are prone to creating a crust. The content of the nutrients in the soil and pH are given in Table 1.

Fertilization with 100 kg ha^−1^ P (as superphosphate) and 150 kg ha^−1^ K (potassium salt) was carried out within basic tillage. Flax was fertilized with different N rates (0, 30, 60 and 90 kg ha^−1^) at different times (urea (46%) and with calcium ammonium nitrate (27% N)). No nitrogen was added in the first trial treatment. In the second fertilization treatment, all nitrogen was added before sowing (30 kg N/ha). In the third fertilization treatment, 30 kg N/ha was added before sowing, and 30 kg in a single fertilizer application was added at the average plant height of 10 cm; in the fourth fertilization treatment, 30 kg was added before sowing, and 30 kg was added at the average plant height of 10 and 20 cm each.

The trials were laid out according to the randomized complete block design (RCBD) with four replications. The main trial plot size was 10 m^2^ (10 rows × 0.1 m row spacing × 10 m length). Sowing was carried out using a plot seeder (Wintersteiger, Ried, Austria. Fibre flax seeding was performed on 31 March 2008, 1 April 2009 and 29 March 2010. Sowing density was 2500 germinable seeds/m^2^. The protocol for investigating the effect of nitrogen on the properties of flax plants and fibres are shown in Figure 1.

### 2.2. Plant Properties

The investigated morphological properties were technical stem length and stem thickness. Pulling by hand was carried out at the yellow–green ripening stage. Plants were pulled from an area of 1 m^2^. Technical stem length was measured from the cotyledon node to the first branch. Stem thickness was determined in the middle of the technical stem length using an electronic micrometre. Flax stems were then placed in tank of water at 30 °C for 4 days under controlled conditions. Afterwards, retting stems were removed from the tank, dried at 60 °C for 30 h and weighed. A scutching machine was used to separate straw (woody matter) from the fibres.

### 2.3. Fibre Properties

Flax fibres are characterized by the following textile-technological properties: length, fineness and tenacity. Fibre length was determined by the measurement of technical fibres method according to HRN ISO 6989 2003. Fineness (HR EN ISO 1973 2021) and tenacity (HR EN ISO 5079 2020) of flax fibres were determined on tensile testers, Vibroscop and Vibrodyn 400 (Lenzing Instruments, Gampern, Austria). Used standards and regulations are adapted for testing technical flax fibres. For this purpose, cogged steal clamps were placed on the standard clamps, and the selected testing speed was 3 mm/min with a load cell capacity of 1500 mg. The selected gauge length of the sample was 5 mm to ensure that all elementary fibres were included in the tested sample during fineness and tenacity testing.

Used standards and regulations were adapted for testing technical flax fibres. Due to the non-homogeneity of flax fibres, the number of measurements was increased from 50 to 100. The measurements of the tested properties of fibres were performed on conditioned samples.

### 2.4. Statistical Analysis

Data of all the properties studied in each location and year were statistically processed by an analysis of variance. Differences between mean values were analysed using Duncan’s multiple range test (DMRT) [28].

## 3. Results and Discussion

Statistically significant differences were recorded among the cultivars for the investigated properties of fibre flax, except for stem thickness in 2008 at Križevci and in 2010 at both locations; fibre length in 2010 at Zagreb; fibre fineness in 2008 at Križevci, 2009 at Zagreb and 2010 at both locations; and fibre tenacity in 2008 at Zagreb and 2010 at Križevci (Table 2 and Table 3).

In addition, statistically significant differences were recorded among different nitrogen rates for technical stem length in 2008 at both locations and in 2009 at Krizevci; stem thickness in 2008 at both locations and in 2009 at Zagreb; fibre length in 2008 and 2009 at both locations and in 2010 at Križevci; fibre fineness in 2009 at Zagreb; and fibre tenacity in 2010 at Krizevci (Table 4 and Table 5). No significant interaction was recorded for any properties or any locations; therefore, interactions were not included in the factors shown here and were not discussed any further. Accordingly, the factors affected the studied properties independently.

The technical stem length should not be shorter than 60 cm, and it is desirable at a length of about 100 cm. The obtained values (Table 2 and Table 3) of technical stem length for all cultivars in 2008 and 2009 were below 60 cm, except in 2010 when they were higher than 60 cm (except for the Viking cultivar) at location Zagreb. At Križevci, the obtained values of technical stem length for all cultivars in all years were higher than 60 cm, except for the Viking cultivar in 2008. The highest technical stem lengths were achieved by the cultivar Agata in 2008 and 2010 at Zagreb and cultivar Viola at Križevci in all three years of investigation.

Slightly lower values than the average, as determined by a group of authors in their previous research [29,30,31], are the result of unfavourable weather conditions during the growth and development of flax (after planting—low temperature with the appearance of frost; excessively high temperatures in May—premature flowering; excessive rainfall in June—unevenly ripening). In addition, the production of fibre flax in sandy soil at Zagreb is unfavourable in the years with unevenly distribution of rainfall. Stem thickness affects fibre quantity and quality, i.e., the fibres fineness depends on the thickness of the stem. A very thin and very thick stem is less valuable for textile use. The thicknesses of the stems at both locations are between 1.19 to 1.78 mm, which is suitable for obtaining fibres of satisfactory quantity and corroborates the research of other authors [29,30,32].

The thickest stem thickness was achieved by cultivar Viola in 2010 at Zagreb and in 2009 at Križevci. At Križevci, importantly, the longest and thickest stem was achieved by the cultivar Viola in 2009.

Fibre length measurement results are in compliance with the measurement of the technical stem length. The longest fibre length, at Zagreb, was achieved by the cultivar Viola in 2008, Agatha in 2009 and Venica in 2010. At Križevci, the longest fibre length was achieved by the cultivar Viola in 2008 and 2010 and Electra in 2009. Significantly, the longest fibre length was achieved by the cultivar Agatha at Zagreb in 2009.

The finest fibres were obtained from cultivars Venica (2008) and Viola (2009 and 2010) at Zagreb, and cultivars Viking (2008 and 2009) and Venica (2010) at Križevci. The strongest fibres were obtained from cultivar Agatha in 2008 and 2010 and cultivar Viola in 2009 at Zagreb. At Križevci, the strongest fibres were achieved by the cultivars Viola (2008), Viking (2009) and Electra (2010). Significantly, the strongest fibres were achieved by cultivar Viola at Zagreb in 2009. From the analysis of fibre fineness and tenacity, it can be concluded that these factors affected the investigated properties independently, which is consistent with other authors [3]. In addition, the obtained results of fibre properties are in accordance with the research of other authors [4,33,34].

The means of the morphological properties of fibre flax, and the physical-mechanical properties of fibres dependent on nitrogen rates, are given in Table 4 for Zagreb and in Table 5 for Križevci for all three years.

The results of previous research on fertilizing flax with nitrogen are different, and depend on the added amounts of nitrogen, application time, soil nitrogen supply and weather conditions. However, the authors found that there was no significant increase in the properties of the flax and obtained fibres when larger amounts of nitrogen (30 kg/ha) were added, and 30 kg/ha of nitrogen is recommended for obtaining an optimal fibre quality [8,15,17,35,36]. According to the obtained values of the investigated properties, there were no significant differences between the variants in which 30, 60 or 90 kg/ha of nitrogen were added. Importantly, only in 2009, at Zagreb, the thickest stem was achieved with 60 kg/ha of nitrogen, and at Krizevci, the longest fibre length was achieved with 60 and 90 kg/ha of nitrogen. In general, by increasing the fertilization from 30 to 60 kg/ha of nitrogen, the values increase slightly, and after the application of 90 kg/ha of nitrogen they decrease.

Fibre length, fineness and tenacity are the most important processing properties and determine the quality and suitability of flax fibres as a textile raw material for yarn and fabric manufacturing. Furthermore, measurements of fibre properties are in compliance with measurements of plant properties.

## 4. Conclusions

This paper presents a systematic and interdisciplinary evaluation from a three-year study on the effect of different amounts of nitrogen on the properties of flax plants and fibres through the valorisation of their morphological (technical stem length, stem thickness) and physical-mechanical properties (length, fineness, tenacity). For the purpose of evaluating the influence of cultivars, location and nitrogen fertilization on plant and fibre properties, a number of parameters were taken into consideration that affect plant growth and development as well as fibre properties.

Based on the analysis of the results, individual conclusions can be made depending on the investigated properties:-Statistically significant differences were recorded among the cultivars for the investigated properties of fibre flax, except for stem thickness in 2008 at Križevci and in 2010 at both locations; fibre length in 2010 at Zagreb; fibre fineness in 2008 at Križevci, 2009 at Zagreb and 2010 at both locations; and fibre tenacity in 2008 at Zagreb and 2010 at Križevci.-Statistically significant differences were recorded among different nitrogen rates for technical stem length in 2008 at both locations and 2009 at Krizevci; stem thickness in 2008 at both locations and 2009 at Zagreb; fibre length in 2008 and 2009 at both locations and in 2010 at Križevci; fibre fineness in 2009 at Zagreb; and fibre tenacity in 2010 at Krizevci.-At Križevci, the longest and thickest stem was achieved by the cultivar Viola in 2009. In addition, the cultivar Viola obtained the strongest fibres at Zagreb in 2009. At Zagreb in 2009, the longest fibre length was achieved by the cultivar Agatha.-Based on the results of the morphological and textile-technological properties of flax, the cultivars Viola and Agatha achieved higher values, and their fibres satisfy the current market demands for their use.-Nitrogen-fertilized variants had longer technical stem lengths and fibre lengths, thicker stems, and slightly coarser and stronger fibres. Since there is no significant increase in the properties of flax and obtained fibres when larger amounts of nitrogen of 30 kg/ha were added, there is no need to add more than 30 kg/ha of nitrogen in the fertilization of flax.

Based on the results of the morphological and textile-technological properties of flax, the cultivars Viola and Agatha achieved higher values at Križevci, and it is not necessary to add more than 30 kg of N/ha.

## Figures and Tables

**Figure 1 polymers-14-00558-f001:**
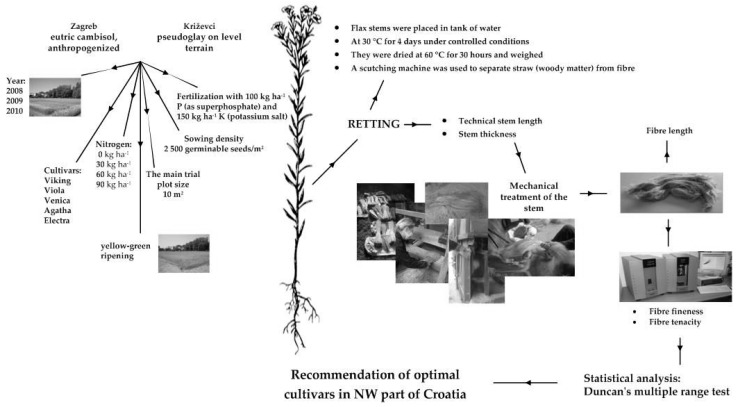
The protocol for investigating the effect of nitrogen on the properties of flax plants and fibres.

**Table 1 polymers-14-00558-t001:** The content of nutrients in the soils and pH.

Year	P_2_O_5_ (mg/100 g)	K_2_O (mg/100 g)	Total Nitrogen (%)	pH (KCl)
Zagreb				
2008	22.6	18.0	0.18	7.21
2009	21.1	25.0	0.15	6.94
2010	19.9	19.9	0.12	6.91
Križevci				
2008	24.2	18.3	0.11	4.59
2009	28.6	17.6	0.10	4.21
2010	20.9	19.4	0.09	5.04

P_2_O_5_, K_2_O—Al-method; Total nitrogen—HRN ISO 13878:2004; pH—HRN ISO 10390:2004.

**Table 2 polymers-14-00558-t002:** Means of morphological properties of fibre flax and physical-mechanical properties of fibres dependent on the cultivars at Zagreb (2008, 2009 and 2010).

Cultivars	Technical Stem Length (cm)	Stem Thickness (mm)	Fibre Length (cm)	Fibre Fineness (dtex)	Fibre Tenacity (cN/tex)
2008					
Viking	49.10 c	1.21 c	33.62 b	35.92 b	73.10 a
Viola	55.39 b	1.26 bc	39.35 a	37.16 ab	75.83 a
Venica	56.04 ab	1.40 ab	35.09 b	34.63 b	73.75 a
Agatha	59.46 a	1.43 a	38.55 a	36.75 ab	76.45 a
Electra	56.18 ab	1.41 ab	35.34 b	39.44 a	73.43 a
2009					
Viking	43.88 b	1.23 c	27.79 d	37.31 a	67.32 b
Viola	51.56 a	1.32 b	31.82 b	36.20 a	73.23 a
Venica	50.00 a	1.38 b	28.46 d	37.34 a	65.33 b
Agatha	51.25 a	1.41 ab	34.27 a	38.15 a	65.74 b
Electra	51.31 a	1.49 a	29.98 c	37.01 a	65.78 b
2010					
Viking	56.31 c	1.42 a	28.81 a	37.62 a	69.67 ab
Viola	67.94 ab	1.53 a	29.96 a	35.51 a	65.37 ab
Venica	62.81 b	1.51 a	30.34 a	35.75 a	69.64 ab
Agatha	69.44 a	1.44 a	28.86 a	36.61 a	72.12 a
Electra	64.25 ab	1.51 a	28.91 a	36.29 a	64.68 b

Values with the same letter are not significant at a level of 5%; as the significance value decreases, the letters become lower since they are in alphabetical order.

**Table 3 polymers-14-00558-t003:** Means of morphological properties of fibre flax and physical-mechanical properties of fibres dependent on the cultivars at Križevci (2008, 2009 and 2010).

Cultivars	Technical Stem Length, cm	Stem Thickness, mm	Fibre Length (cm)	Fibre Fineness (dtex)	Fibre Tenacity (cN/tex)
2008					
Viking	57.00 b	1.33 a	33.55 c	36.44 a	73.19 b
Viola	68.43 a	1.42 a	43.39 a	36.67 a	84.26 a
Venica	65.93 a	1.40 a	38.86 b	37.82 a	79.13 ab
Agatha	67.37 a	1.43 a	41.46 ab	37.49 a	84.01 a
Electra	66.56 a	1.34 a	39.13 b	38.31 a	83.75 a
2009					
Viking	66.49 d	1.33 b	46.55 d	33.85 c	79.42 a
Viola	77.17 a	1.78 a	52.51 ab	35.49 abc	77.52 ab
Venica	69.16 cd	1.29 b	50.39 bc	34.70 bc	78.85 a
Agatha	72.56 b	1.43 b	47.23 cd	37.29 a	72.27 b
Electra	70.43 bc	1.41 b	55.13 a	36.46 ab	78.08 a
2010					
Viking	63.61 b	1.26 a	32.53 ab	38.76 a	70.44 a
Viola	75.42 a	1.28 a	35.02 a	38.46 a	67.98 a
Venica	66.59 a	1.19 a	31.22 b	36.09 a	69.86 a
Agatha	72.82 a	1.27 a	33.80 ab	36.98 a	69.22 a
Electra	73.63 a	1.26 a	31.02 b	36.19 a	70.85 a

Values with the same letter are not significant at a level of 5%; as the significance value decreases, the letters become lower since they are in alphabetical order.

**Table 4 polymers-14-00558-t004:** Means of morphological properties of fibre flax and physical-mechanical properties of fibres dependent on nitrogen rates at Zagreb (2008, 2009 and 2010).

Nitrogen	Technical Stem Length, cm	Stem Thickness, mm	Fibre Length (cm)	Fibre Fineness (dtex)	Fibre Tenacity (cN/tex)
2008					
0	52.79 b	1.26 b	34.41 b	36.16 a	77.32 a
30	56.36 a	1.39 a	37.47 a	36.52 a	74.02 a
60	56.57 a	1.41 a	36.69 ab	37.27 a	76.00 a
90	54.84 a	1.30 a	35.99 ab	37.16 a	70.71 a
2009					
0	47.80 a	1.23 c	31.26 a	36.88 ab	64.99 a
30	50.65 a	1.37 b	31.79 a	38.44 a	68.20 a
60	50.75 a	1.47 a	29.94 b	38.03 ab	68.58 a
90	49.20 a	1.38 b	28.87 b	34.60 b	68.15 a
2010					
0	61.85 a	1.37 a	29.61 a	35.50 a	71.72 a
30	64.05 a	1.49 a	28.74 a	35.48 a	68.20 a
60	66.50 a	1.51 a	30.20 a	36.79 a	68.58 a
90	64.20 a	1.55 a	28.97 a	37.64 a	68.15 a

Values with the same letter are not significant at a level of 5%; as the significance value decreases, the letters become lower since they are in alphabetical order.

**Table 5 polymers-14-00558-t005:** Means of morphological properties of fibre flax and physical-mechanical properties of fibres dependent on nitrogen rates at Križevci (2008, 2009 and 2010).

Nitrogen	Technical Stem Length, cm	Stem Thickness, mm	Fibre Length (cm)	Fibre Fineness (dtex)	Fibre Tenacity (cN/tex)
2008					
0	61.70 b	1.19 b	37.47 b	37.09 a	80.31 a
30	66.85 ab	1.44 a	39.35 ab	37.17 a	81.22 a
60	67.45 a	1.46 a	40.26 a	37.20 a	78.79 a
90	64.30 ab	1.39 a	39.54 ab	38.02 a	83.16 a
2009					
0	69.16 b	1.30 a	46.84 b	34.98 a	77.54 a
30	71.72 ab	1.49 a	48.71 b	36.40 a	77.28 a
60	73.08 a	1.54 a	52.69 a	35.64 a	77.15 a
90	70.70 ab	1.47 a	53.22 a	35.22 a	76.95 a
2010					
0	64.61 a	1.20 a	32.27 a	36.64 a	69.85 a
30	71.96 a	1.21 a	32.71 a	36.06 a	72.01 a
60	73.64 a	1.27 a	34.22 a	38.96 a	71.65 a
90	71.44 a	1.30 a	31.38 b	37.54 a	65.17 b

Values with the same letter are not significant at a level of 5%; as the significance value decreases, the letters become lower since they are in alphabetical order.

## Data Availability

Data is available upon request.

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
