# Peer review of "Effect of Nitrogen on the Properties of Flax (*Linum usitatissimum* L.) Plants and Fibres"

_polymers, 2022, doi:10.3390/polym14030558_

Round 1
Reviewer 1 Report
The article should be restructured. The introduction is too broad in comparison to results and conclusions, which should be the essence of the text.
Lines 118 to 140 are a conclusion but have been added in the introduction. The Materials and Methods section should be divided into subsections describing each point of the methodology. In addition, a map of the sampling should be added.
Duncan's Multiple Range Test (DMRT) is a post hoc test, so it should be preceded by an ANOVA. the authors have not fully described the methodology of the statistical analysis. Furthermore, on what basis was this analysis carried out when the tables do not show the standard deviations? Moreover, the results of the statistical analysis should be written in superscript. The authors did not provide information about the level of significance (p-value) under the tables and in the methodology.
The discussion could be more elaborated and supported by the novel literature, for example:
Siegien, I., Fiłoc, M., Staszak, A., & Ciereszko, I. (2021). Cyanogenic glycosides can function as nitrogen reservoir for flax plants cultured under N-deficient conditions. Plant, Soil and Environment, 67(4), 245-253.
Kakabouki, I., Mavroeidis, A., Tataridas, A., Roussis, I., Katsenios, N., Efthimiadou, A., ... & Bilalis, D. (2021). Reintroducing Flax (Linum usitatissimum L.) to the Mediterranean Basin: The Importance of Nitrogen Fertilization. Plants, 10(9), 1758
"Means of morphological properties of fibre flax and physical-mechanical properties of fibres in dependence of nitrogen rates at Zagreb" - Did the authors count the correlation coefficient?
There are editorial errors in the conclusions section. There are no fixed elements at the end, such as author contributions.
The manuscript is not suitable for republication in its current version
Author Response
I thank you for your constructive comments and I believe it will improve the proposed manuscript.

Reviewer 2 Report
The results presented are interesting but the analysis and the form of the presentation must be greatly improved to merit publication in a scientific journal.
The results are displayed in simple tables. Graphics would provide a view of the results. Radar or spider type charts might be a good solution to easily compare the results.
The authors announce results based on statistical work, but only the average value is given. Other statistical information deserves to be exploited.
A cross-analysis, of the ANOVA type, between the different measured values would be worth doing to highlight any correlations between the properties themselves, the properties and the growth conditions.
Author Response

(The authors gave the same response as above.)

Reviewer 3 Report
The authors investigated the effect of nitrogen content on the properties of 5 varieties of flax fibers. The work is an extension of their previous study, and the results are intriguing. The following are my comments/suggestions to improve the work further.
Line 43: The statement is too direct. There are several studies that investigated the link between variety, growing conditions, and properties. Authors should consider citing the following articles since these works are already known in the domain of flax fibers which also speak about the variation of mechanical properties.
https://doi.org/10.1016/j.indcrop.2016.11.062
https://doi.org/10.1016/j.indcrop.2021.113736
https://doi.org/10.1016/j.indcrop.2016.12.028
https://doi.org/10.1016/j.indcrop.2019.111710
Line 67: Authors can be more specific by stating what they mean by quality.
Line https://doi.org/10.1016/j.indcrop.2014.07.015
Line 97: Stem fibers or bast fibers?
Line 108: any references for coarser fibers?
Line 182: What do the authors mean by single fibers? Elementary or technical? Please clarify.
Line 184: What is the load cell capacity?
Line 187: How were the technical fibers separated? Manually? And why were the technical fibers selected over elementary fibers?
Line 190: What was this 'increased' number?
Table 2: The units should be presented in parentheses. For example length of fibers (cm)
Also, the titles can be shortened, remove 'of' and mention 'Fiber length' 'Fiber tenacity.'
Correct the spelling 'steam'--> stem
The statistical indicators (abcd) should be in superscript. It is easier to read.
Also, please add the standard deviations.
Throughout the document 'Steam'--> 'stem.'
Table 4: the first column is not Cultivars. Its nitrogen content.
And which cultivar does this table belong to?
And what about the other four cultivars? Was the variation statistically insignificant?
Table 5: Same questions as above.
Also, add standard deviations for each value
Author Response

(The authors gave the same response as above.)

Round 2
Reviewer 1 Report
In my opinion, the work has been corrected and can be published
Reviewer 2 Report
Accepted in present form